# Manganese Stress Tolerance Depends on Yap1 and Stress-Activated MAP Kinases

**DOI:** 10.3390/ijms232415706

**Published:** 2022-12-11

**Authors:** Inés G. de Oya, Elena Jiménez-Gutiérrez, Hélène Gaillard, María Molina, Humberto Martín, Ralf Erik Wellinger

**Affiliations:** 1Centro Andaluz de Biología Molecular y Medicina Regenerativa, Universidad de Sevilla, Avda. Américo Vespucio s/n, 41092 Sevilla, Spain; 2Departamento de Genética, Universidad de Sevilla, Avda. Reina Mercedes 6, 41012 Sevilla, Spain; 3Departamento de Microbiología y Parasitología, Facultad de Farmacia, Universidad Complutense de Madrid, 28040 Madrid, Spain; 4Instituto Ramón y Cajal de Investigaciones Sanitarias (IRYCIS), 28034 Madrid, Spain

**Keywords:** manganese, Pmr1, stress response, Yap1, MAP kinases

## Abstract

Understanding which intracellular signaling pathways are activated by manganese stress is crucial to decipher how metal overload compromise cellular integrity. Here, we unveil a role for oxidative and cell wall stress signaling in the response to manganese stress in yeast. We find that the oxidative stress transcription factor Yap1 protects cells against manganese toxicity. Conversely, extracellular manganese addition causes a rapid decay in Yap1 protein levels. In addition, manganese stress activates the MAPKs Hog1 and Slt2 (Mpk1) and leads to an up-regulation of the Slt2 downstream transcription factor target Rlm1. Importantly, Yap1 and Slt2 are both required to protect cells from oxidative stress in mutants impaired in manganese detoxification. Under such circumstances, Slt2 activation is enhanced upon Yap1 depletion suggesting an interplay between different stress signaling nodes to optimize cellular stress responses and manganese tolerance.

## 1. Introduction

Manganese is an essential, redox active metal of medical relevance. It serves as a metal co-factor for enzymes located in all cellular compartments and environmental exposure to manganese has been associated with the development of Parkinson-like neurological symptoms in human [1]. Budding yeast has been pivotal to identify genetic mutations affecting manganese metabolism and cellular availability. Manganese is sequestered by the vacuole or the Golgi, and Golgi manganese detoxification is mainly provided by the P-type Ca^2+^/Mn^2+^ ATPase Pmr1 [2]. The Pmr1 protein can be replaced by its human homolog SPCA1 encoded by the *ATP2C1* gene [3] and mutations affecting calcium and/or manganese transport activities of SPCA1 are linked to Hailey-Hailey disease (HHD, [4]). Noteworthy, yeast cells lacking Pmr1 are sensitive to peroxide (H_2_O_2_; [5]) but at the same time, the absence of Pmr1 suppress oxygen-mediated phenotypes of superoxide dismutase 1 (Sod1) mutants [6].

To adapt to diverse extracellular stimuli and environmental changes, cells activate stress response pathways known to operate through sequential phosphorylation events that are termed protein kinase cascades. The stress-activated MAP kinases (SAPK) signaling cascades are evolutionary conserved prototypes of this kind of stress response signaling pathways. In yeast, the Hog1 SAPK plays a key role in reprogramming the gene expression pattern required for cell survival upon osmostress [7], while the Slt2 SAPK is the downstream kinase of the so-called cell wall integrity (CWI) pathway [8]. The human Hog1 paralog p38 plays a critical role in adaptive responses to environmental stress [9]. Slt2 is a functional homolog of human extracellular signal-regulated kinase 5 (*ERK5*), a MAPK that is activated in response to growth factors as well as physical and chemical stresses [10]. Together, Hog1 and Slt2 are needed to coordinate metabolic needs with cell cycle progression, therewith contributing to maintain genetic stability in yeast [11,12].

In an adaptive response to hydrogen peroxide (H_2_O_2_) treatment-mediated oxidative stress, the transcription factor yeast activator (AP1-like) protein (Yap1) is activated and translocates into the nucleus to induce the expression of a protective transcriptional program [13,14]. During metal and heat-shock induced oxidative stress, Yap1 transcriptionally regulates the expression of *GSH1*, which encodes the first enzyme (γ-glutamylcysteine synthetase) involved in glutathione biosynthesis, leading to an increase in intracellular glutathione level [15,16].

A growing body of evidence provides a link between the loss of P-type Golgi Ca^2+^/Mn^2+^ ATPases and genome instability in yeast and human. Loss of Pmr1 function leads to enhanced sensibility to a variety of DNA damaging agents, DNA damage formation and telomere shortening [5,17,18], while depletion of *ATP2C1* in human cells was shown to boost the formation of reactive oxygen species (ROS) and to down-regulate the expression of DNA damage response (DDR) genes [19]. Finally, a genetic screen in *Kluyveromyces lactis* identified the Glutathione S-transferase ϴ-subunit (*GTT1*) as an oxidative stress suppressor in cells lacking Pmr1 [20], thereby providing a link between manganese and oxidative stress response.

These findings prompted us to further investigate stress response pathways related to oxidative damage and to determine if they contribute to manganese stress resistance in yeast. Here, we reveal that impaired manganese homeostasis leads to oxidative stress and that cellular tolerance to MnCl_2_ requires the Yap1 transcription factor, although MnCl_2_ addition stimulates Yap1 decay. Compromised manganese detoxification leads to a constitutive activation of the CWI effector MAPK Slt2, and Slt2 activation is further stimulated in the absence of Yap1. The activation of different stress signaling nodes pinpoints to a multifaceted impact of manganese on cellular metabolism that requires the concerted action of signaling kinases and transcription factors for manganese and oxidative stress tolerance.

## 2. Results

### 2.1. Impaired Manganese Homeostasis Leads to Oxidative Stress

Several observations relate Golgi Ca^2+^/Mn^2+^ transport to oxidative stress. The viability of yeast cells lacking a functional Pmr1 Golgi Ca^2+^/Mn^2+^ ATPase is compromised upon treatment with H_2_O_2_ [5] and oxidative stress and Notch1 activation were increased upon inactivation of the *PMR1* homologue *ATP2C1* in human cultured keratinocytes [19]. In yeast, several pathways participate in oxidative stress signaling, including the activation of the AP-1-like transcription factor Yap1 [21], and the SAPKs Hog1 and Slt2 [22,23] (see Figure 1A).

First, we assessed cell viability and ROS formation in *pmr1Δ* mutants using the fluorescent dyes propidium iodide (PI) and dihydroethidium (DHE), respectively (Figure 1B). As compared to wild-type cells (Wt), we found that loss of Pmr1 led to a considerable increase in the fluorescence signal with either dyes indicative of loss of plasma membrane selective permeability or increased ROS formation in *pmr1Δ* mutants. The later finding is in concordance with previous observations showing that DHE fluorescence is increased in *pmr1Δ* mutants [24]. Next, we wondered if oxidative stress tolerance in cells lacking Pmr1 may depend on Yap1, a transcription factor needed to activate the transcription of antioxidant genes in response to H_2_O_2_ [25,26]. In our strain background, no differences between Wt and *pmr1Δ* cells were observed in growth assays with H_2_O_2_ supplemented media (Figure 1C). However, mutant cells lacking Yap1 alone were very sensitive to H_2_O_2_ and *pmr1Δ yap1Δ* double mutants became even more H_2_O_2_-sensitive than *yap1Δ* single mutants, supporting the previous notion that *pmr1Δ* mutants are prone to oxidative stress. We then analyzed whether H_2_O_2_ induced Yap1 posttranslational modifications, observed as a shift in the electrophoretic mobility (Figure 1D). The cells were treated for up to 1 h with 400 µM H_2_O_2_ and protein samples were taken at various time points. In accordance with previous reports [27], Western blotting analysis revealed a rapid shift in Yap1 migration (7.5 min after treatment), suggesting an increase in Yap1 phosphorylation. While the kinetics of Yap1 protein modification was very similar in Wt and *pmr1Δ* cells, the initial levels of Yap1 protein appeared to be increased in cells lacking Pmr1 (Figure 1D). In addition, we monitored intracellular Yap1-GFP protein localization to assess if the cytoplasmic-nuclear Yap1 transition in response to H_2_O_2_ treatment is altered in *pmr1Δ* mutants (Figure 1E). Fluorescence microscopy analysis showed that this was not the case as the Yap1-GFP protein localization was mainly cytoplasmic in untreated Wt and *pmr1Δ* mutant cells but became principally nuclear in both cell types shortly after H_2_O_2_ treatment.

### 2.2. Manganese-Dependent Yap1 Decay Is Calcineurin B-Independent

Our finding that Yap1 is needed for oxidative stress resistance in the absence of Pmr1 led us to investigate whether or not Yap1 is required for growth on MnCl_2_ supplemented medium. To do so, we performed a cell growth analysis of cells lacking Pmr1 and/or Yap1 in growth medium supplemented with 1, 2.5 or 5 mM MnCl_2_ (Figure 2A). As expected, the growth of *pmr1Δ* mutant cells was strongly inhibited by MnCl_2_ addition. MnCl_2_ addition also impaired the growth of *yap1Δ* mutants as compared to Wt control cells. We therefore wondered if MnCl_2_ treatment would have an impact on Yap1 protein levels. Consequently, we performed Western blotting analyses of Wt and *pmr1Δ* cells expressing a Yap1-GFP protein upon treatment with MnCl_2_ (Figure 2B). MnCl_2_ treatment caused a significant reduction of Yap1-GFP protein levels both in Wt and *pmr1Δ* cells after the addition of 10 mM MnCl_2_. MnCl_2_ treatment did not alter the Yap1-GFP migration pattern suggesting that MnCl_2_ is not associated with Yap1 posttranslational modifications. In order to assess if the reduction of Yap1 protein levels is associated with a change in its subcellular localization, we monitored Yap1-GFP distribution by fluorescence microscopy (Figure 2C). However, in contrast to the nuclear Yap1-GFP accumulation observed in H_2_O_2_ treated cells, Yap1-GFP did not accumulate in the nucleus upon MnCl_2_ addition.

In order to distinguish whether Yap1 was down-regulated at the translational or post-translational level, we examined the effect of MnCl_2_ on the stability of Yap1 in the presence of protein synthesis inhibitor cycloheximide (CHX, Figure 3A). Notably, the decrease of Yap1-GFP protein levels was very similar in the presence of MnCl_2_ or CHX. No additive effect on Yap1 protein decay was observed in the presence of both MnCl_2_ and CHX suggesting that MnCl_2_ interferes with Yap1 protein synthesis. We next wondered if MnCl_2_-induced Yap1 decay is linked to the previously described CaCl_2_-dependent Yap1 decay due to activation of the calcium/calmodulin regulated serine/threonine protein phosphatase calcineurin [28]. Calcineurin is a heterodimeric enzyme comprising of a catalytic A and a regulatory B subunit, and is required for Crz1 transcription factor dependent gene expression, ion homeostasis and viability in yeast [29]. By taking advantage of calcineurin B (*cnb1Δ*) mutant cells we confirmed that the calcineurin B subunit is required for MnCl_2_ tolerance, in accordance with impaired activation of Crz1 target genes related to ion homeostasis (Figure 3B). However, in our strain background the CHX-induced Yap1 decay was barely reduced in *cnb1Δ* mutant cells, and MnCl_2_-dependent Yap1 decay was similar in Wt cells as compared to *cnb1Δ* mutants (Figure 3C). Further experimental settings will be needed to determine all the aspects of calcineurin-independent, MnCl_2_-driven Yap1-GFP decay.

### 2.3. Pmr1 Depletion Leads to Constitutive Activation of the Slt2 MAP Kinase Pathway

MAPK cascades are among the major pathways by which extracellular stimuli are transduced into intracellular responses in eukaryotic cells. The yeast cell wall integrity (CWI) pathway operates through the sequential activation of protein kinases Bck1, Mkk1/2 and the effector MAPK Slt2 (outlined in Figure 4A). Interestingly, CWI pathway activation occurs in cells exposed to oxidative stress inducing agents, such as H_2_O_2_ [23]. In addition to Slt2, fungal resistance to a variety of stresses depends on the MAPK Hog1 [22]. Since ROS are increased in cells lacking Pmr1, we asked whether Slt2 or Hog1 are activated in *pmr1Δ* mutants. We therefore analyzed the levels of phosphorylated Slt2 and Hog1 in Wt and *pmr1Δ* mutant cells (Figure 4B). In contrast to Hog1, the levels of Slt2 and phosphorylated Slt2 were enhanced in the absence of Pmr1 indicating a constitutive activation of the Slt2 MAPK pathway in this strain. Thus, we addressed whether extracellular MnCl_2_ addition would be sufficient to stimulate Slt2 activation (Figure 4C). Wt and *pmr1Δ* mutants were grown for 1 h in the presence of increasing amounts of MnCl_2_ prior to protein extraction. Subsequent immunoblotting revealed a MnCl_2_ concentration-dependent increase in Slt2 and Hog1 phosphorylation. However, MnCl_2_ concentration-dependent phosphorylation was more evident for Hog1 and we did not observe a difference in Hog1 activation in *pmr1Δ* mutants as compared to Wt cells. To verify that the constitutive Slt2 phosphorylation observed in *pmr1Δ* mutants depends upon its upstream kinases Mkk1 and Mkk2, we compared MnCl_2_- and Congo red (CR)-induced Slt2 phosphorylation in cells devoid of these kinases (Figure 4D). Notably, the phosphorylated Slt2 signal was absent in *mkk1Δ mkk2Δ* (*mkk1,2Δ*) double as well as in *pmr1Δ mkk1Δ mkk2Δ* triple mutants excluding the possibility that MnCl_2_ could drive Slt2 phosphorylation by non-canonical kinases. Moreover, Hog1 phosphorylation was not affected by the concomitant absence of Mkk1 and Mkk2, as expected for Slt2-independent Hog1 phosphorylation in *pmr1Δ* mutants. These findings suggest that Slt2 phosphorylation is a hallmark of Pmr1 deficiency.

### 2.4. The Slt2 Target Rlm1 Is Up-Regulated in Cells Lacking Pmr1

*SLT2* expression is known to be subjected to a feedback mechanism mediated by the CWI pathway specific transcription factor Rlm1 [30]. The observation that Slt2 protein levels are increased in *pmr1Δ* mutants (Figure 4) suggests that Rlm1 activates *SLT2* gene expression. To test for this possibility, we generated *pmr1Δ rlm1Δ* double mutants and analyzed the levels of phosphorylated and total Slt2 (Figure 5A). The total Slt2 protein signal was reduced while phosphorylated Slt2 levels remained high in cells lacking Pmr1 and Rlm1. This finding is consistent with a model in which phosphorylated Slt2 activates Rlm1 to promote transcription of the *SLT2* gene. We therefore took advantage of previously published microarray data [17], to figure out whether the expression of further Rlm1 target genes was increased in *pmr1Δ* mutants. This was indeed the case for a number of known Rlm1 target genes, including the *SLT2* paralog *KDX1* (*MLP1*), which was highly expressed in the absence of Pmr1 (Figure 5B).

We performed a Western blotting analysis and confirmed that Kdx1 protein levels are highly increased in *pmr1Δ* mutants (Figure 5C). Kdx1 has been shown to interact with Rlm1 to drive the expression of stress responsive genes such as *RCK1*, which codes for a protein kinase involved in oxidative stress response [31]. It is noteworthy that Rck1 overexpression has been proposed to modulate Hog1 and Slt2 activation [32]. Next, we investigated if cells lacking Rlm1 and/or Kdx1 become more sensitive to H_2_O_2_ or MnCl_2_ (Figure 5D). As compared to Wt and *pmr1Δ* mutants, colony formation efficiency was similar in the presence or absence of Rlm1 and/or Kdx1, suggesting that these factors are dispensable for manganese and oxidative stress resistance.

### 2.5. Slt2 Contributes to Oxidative Stress Resistance of pmr1Δ Mutants

Having excluded a role for Rlm1 and Kdx1 in the oxidative stress resistance of *pmr1Δ* mutants, we strove to assess the impact of Slt2 itself on oxidative stress resistance. Yeast cells have been previously shown to become highly sensitive to H_2_O_2_ in the absence of Slt2 [23,33] and we therefore analyzed colony formation in the presence of H_2_O_2_ (Figure 6A). Notably, we did not observe a difference between the growth of Wt and *slt2Δ* mutants in H_2_O_2_ supplemented medium, probably because different strain backgrounds were used in our and previous studies. Importantly, *pmr1Δ slt2Δ* double mutants were much more sensitive to H_2_O_2_ than Wt, *pmr1Δ* or *slt2Δ* single mutant cells, providing evidence that Slt2 is required to withstand oxidative stress in the absence of Pmr1. The concomitant absence of Pmr1 and Yap1 resulted in further increased Slt2 phosphorylation compared to the *pmr1Δ* single mutant (Figure 6B). These results suggest that manganese stress signalling is channeled into the CWI pathway in the absence of Yap1 and led us to assess whether oxidative stress is responsible for Slt2 phosphorylation. To this end, cells were treated with the antioxidant N-acetyl cystein [4] to see if Slt2 phosphorylation can be diminished (Figure 6C). This was indeed the case, as N-acetyl cystein treatment strongly reduced Slt2 phosphorylation of *pmr1Δ* simple and *pmr1Δ yap1Δ* double mutants to similar levels. Since composite docking sites confer substrate recognition by both calcineurin and MAP kinases [34], we wondered if Slt2 regulates Yap1 protein levels upon MnCl_2_ exposure. Therefore, cells lacking Pmr1 and/or Slt2 were grown for 1 h in MnCl_2_ supplemented medium and Yap1-GFP signals analyzed by Western blotting (Appendix A). However, MnCl_2_ leads to a similar drop in Yap1 protein levels in the absence of Pmr1 and/or Slt2 and, thus, these proteins are not involved in Yap1 stability or synthesis. Taken together, our results define a relevant role for Slt2 and Yap1 in the response to manganese stress and open the possibility of an additional mechanism involved in the regulation of Yap1 transcription, translation or protein stability.

## 3. Discussion

### 3.1. Yap1 Has a Hitherto Unknown Role in Manganese Stress Response

Manganese is a redox-active metal that can mimic superoxide dismutases by catalyzing the decomposition of O_2_•− to H_2_O_2_ and O_2_ in vitro [35]. The same effect can be achieved in vivo as impaired manganese detoxification in cells devoid of Pmr1 can bypass the lack of Sod1 [6]. If intracellular manganese stimulates H_2_O_2_ formation, intracellular manganese overload upon Pmr1 depletion would provide an explanation for why mutant cells suffer from increased ROS formation. Our findings corroborate the previously observed increase in ROS levels in *ATP2C1* depleted human keratinocytes [19]. We find that Yap1 protein levels are increased in cells lacking Pmr1. However, there are substantial differences between oxidative stress activation of the mammalian stress transcription factor Nrf2 as compared to yeast Yap1, because a complex stress-sensing system mechanism is needed for Yap1 nuclear retention including the assistance of glutathione peroxidase 3 (Gpx3) [27,36,37]. Regarding this unique activation mechanism, the Yap1 system might stem from a different evolutionary origin than the Keap1-Nrf2 system (reviewed in [38]).

To our knowledge, the impact of manganese stress on Yap1 activation has not been thoroughly explored. Interestingly, we find that Yap1 is needed for MnCl_2_ resistance, although MnCl_2_ addition leads to a rapid and dramatic drop in Yap1 protein levels (see Figure 2 and Figure 3). The nuclear accumulation of Yap1 upon oxidant challenge or due to impaired nuclear Yap1 export promotes its proteolytic degradation in a E3 Ubiquitin Ligase Not4-dependent manner [39]. MnCl_2_ may stimulate cytoplasmic to nuclear Yap1 turnover, but there is no evidence of a MnCl_2_-dependent change in Yap1 localization suggesting another mechanistic cause for MnCl_2_-driven Yap1 decay. It is possible that manganese interferes with Yap1 translation as MnCl_2_ was found to reduce total rRNA levels in a dose-dependent manner and to alter overall ribosome profiling [40], or one might even speculate that the 5′-UTR of *YAP1* mRNA can form a riboswitch-like structure [41], whose interaction with manganese modulates *YAP1* gene expression or mRNA stability. Yap1 has been shown to be regulated post-translationally in a calcineurin B-dependent manner [28], opening the possibility that MnCl_2_ could alter the enzymatic activity of calcineurin to promote Yap1 decay [42]. Calcineurin B-independent Yap1 down-regulation upon H_2_O_2_ [39] or MnCl_2_ addition suggests that manganese could drive the modulation of other phosphatase or kinase activities that promote Yap1 decay. Along this line, we recently found that manganese stimulates the enzymatic activity of the TORC1 complex in vitro and in vivo [43]. In any case, although calcineurin does not mediate MnCl_2_-triggered Yap1 decay this phosphatase is needed to activate the stress response transcription factor Crz1. Calcineurin-driven Crz1 dephosphorylation is required for nuclear Crz1 translocation, and impaired Crz1 activation in *cna1Δ*/*cnb1Δ* mutants renders these cells sensitive to manganese or arsenic stress [2,44,45]. Thus, it is likely that the balanced action of Yap1 and Crz1 activation is required for manganese stress tolerance.

### 3.2. SAPKs Activation Is a Read-Out for Manganese Stress

Deciphering the activation of different signaling pathways is important to define a manganese stress signature. The depletion of Pmr1 did not stimulate phosphorylation of the osmotic stress effector Hog1. In contrast, extracellular MnCl_2_ did activate Hog1 phosphorylation in a dose-dependent manner. Notably, MnCl_2_ addition results in a transient and similar induction of Hog1 phosphorylation in Wt and *pmr1Δ* mutants. Thus, Hog1 is required for an adaptive response to MnCl_2_ extracellular addition.

In addition to Hog1, extracellular MnCl_2_ supplementation promotes Slt2 phosphorylation in Wt and *pmr1Δ* mutants in a dose-dependent manner (see Figure 4). The total Slt2 levels and Slt2 phosphorylation were steadily increased in cells lacking Pmr1. The increase in Slt2 protein levels was dependent on the Slt2 downstream target Rlm1, a transcription factor that is part of a feedback loop to enhance *SLT2* expression [30]. A hallmark for impaired cell wall integrity is the hypersensitivity to the cell wall damaging agents, a phenotype associated with the loss of Pmr1 function [46]. However, Slt2 activation could be a consequence of Golgi manganese depletion and consequential protein glycosylation and trafficking defects in the absence of Pmr1 [47].

Rlm1 has been shown to interact with the Slt2 paralog Kdx1 (Mlp1) to drive the expression of protein kinase Rck1 involved in oxidative stress response [31]. Indeed, we confirmed that Kdx1 protein levels were up-regulated in cells lacking Pmr1. However, genetic analysis failed to detect a growth defect if cells devoid of Pmr1 and/or Rml1/Kdx1 were grown in the presence of H_2_O_2_. Thus, it is unlikely that Rlm1/Kdx1 have a significant role in oxidative stress tolerance linked to manganese.

### 3.3. Activation of the CWI Stress Response in the Absence of Yap1

Corroborating previous observations that Slt2 protects cells from oxidative stress, a concomitant lack of Pmr1 and Slt2 renders cells hypersensitive to H_2_O_2_ treatment. Slt2 has been shown to promote transcription activation and the elongation of stress-induced genes by catalytic and non-catalytic mechanisms [48,49,50].

It is likely that in the absence of Yap1, the oxidative stress response is channeled into the CWI pathway. The molecular bases of Slt2 activation in the absence of Yap1 remains to be explored, but Slt2 activation may result from impaired Slt2 dephosphorylation, as has occurred with genotoxic stress [51]. It will be interesting to determine if reduced phosphatase activity is linked to enhanced Slt2 phosphorylation in cells lacking Pmr1, and if Slt2 indeed drives the transcriptional activation of oxidative stress-induced genes.

### 3.4. Concluding Remarks and Perspectives

Each kind of stress requires an adequate response to optimize cell survival. How stress signaling networks manage to crosstalk with each other is not well understood, but mechanistic evidence have been provided on how oxidative stress inhibits pheromone signaling [14]. Here, we report the results on the stress response signature of cells supplemented with extracellular manganese and/or lacking the Mn^2+^/Ca^2+^ Golgi transporter Pmr1. Thereby, we reveal the need for concomitant activation of various stress signaling pathways including Yap1 and SAPKs driven signaling as outlined in Figure 6D. A yet unanswered question is how stress signaling is channeled into the MAPK Slt2 in the absence of Yap1. In addition to the hitherto unknown role of Yap1 in manganese tolerance, we find that manganese induces a rapid reduction of Yap1 protein levels. However, the molecular bases of this manganese-driven Yap1 decay still remain to be explored in detail.

## 4. Materials and Methods

### 4.1. Yeast Strains, Plasmids, and Growth Conditions

Yeast strains and plasmids used in this study are listed in Table 1 and Table 2, respectively. Yeast transformants were grown in liquid or solid, adenine supplemented YPD medium (YPAD). Yeast mutants were generated by according to standard protocols by direct gene knock-out or N-terminal tagging.

### 4.2. Drug Sensitivity Assays

Yeast cells were adjusted in concentration to an initial A_600_ of 0.5, and were then serially diluted 1:10 and spotted onto plates supplemented with MnCl_2_ (Sigma, St. Louis, MO, USA; 244589), or hydrogen peroxide (Alfa Aesar, Heysham, UK; L14000) at the indicated concentrations. The plates were incubated at 26 °C over the course of 2–3 days.

### 4.3. Yap1-GFP Localization

The cells were grown to exponential phase prior to fixation with 2.8% paraformaldehyde. Bright field images (DM-6000B, Leica, Wetzlar, Germany) were obtained at a 100× magnification, respectively. Fluorescence was detected using standard filters for mCherry (595 nm excitation/645–675 nm emission) and GFP (480 nm excitation/527 nm emission) and a digital charge-coupled device camera (DFC350, Leica) and pictures were processed with LAS AF (Leica).

### 4.4. Preparation of Yeast Extracts and Immunoblot Analysis

Yeast cells were grown overnight in the appropriate medium, diluted to A_600_ = 0.3 and grown for 3 h at 26 °C prior to addition of MnCl_2_, H_2_O_2_ or Cycloheximide (CHX). Protein extracts were isolated as previously described [43], separated by sodium dodecyl sulfate-polyacrylamide gel electrophoresis (SDS/PAGE), using 8% (*w*/*v*) polyacrylamide (37.5:1), and transferred to nitrocellulose membranes (Hybond, GE Healthcare, Amersham, UK) according to standard protocols. The Slt2 TEY phosphorylation site was marked a rabbit anti-phospho-p44/42 antibody (Thr-202/Tyr-204; Cell Signaling, Danvers, MA, USA, catalog# 9101), Hog1 phosphorylation with a p38 MAPK Thr180/182 antibody (Cell Signaling, catalog# 9101), Slt2 with an anti-Slt2 mouse antibody (Santa Cruz, Dallas, TX, USA, catalog# sc-374440), GFP with a anti-GFP mouse antibody (JL-8, Clontech, Shiga, Japan, catalog# 632381), G6DPH by an anti-G6DPH rabbit antibody (Sigma, catalog# A9521) and Pgk1 with an anti-Pgk1 mouse antibody (Invitrogen, Waltham, MA, USA, catalog# 459250) prior to inmuno-detection with a secondary antibody. Western blot images were acquired using a Chemidoc™ MP system and Image Lab software (Bio-Rad, Hercules, CA, USA). All immunoblots were reproduced at least twice in independent experiments with representative images shown.

### 4.5. Microarray Analysis

Microarray Analysis—Gene expression profiles were determined by using the “3′-expression microarray” technology by Affymetrix platform at the Genomics Unit of CABIMER (Seville, Spain) as described previously [5,17,18]. The microarray data can be derived from the GEO database using the identifier GSE29420.

### 4.6. ROS Detection Assay

For the detection of oxygen free radicals (ROS) and variations in the mitochondrial membrane potential or cell viability, yeast cells were grown on YPD and cultured as usual, and 2.5 µg/mL of dihydroethidium (DHE) or 5 µg/mL of propidium iodide (PI), respectively, were added to 1 mL of each sample and incubated at 24 °C for 5 and 30 min. Next, the samples were diluted 1:10 in PBS and analyzed on a FACScalibur flow cytometer (Becton Dickinson, Franklin Lakes, NJ, USA). The data were analyzed with FlowJo software (Becton Dickinson).

## Figures and Tables

**Figure 1 ijms-23-15706-f001:**
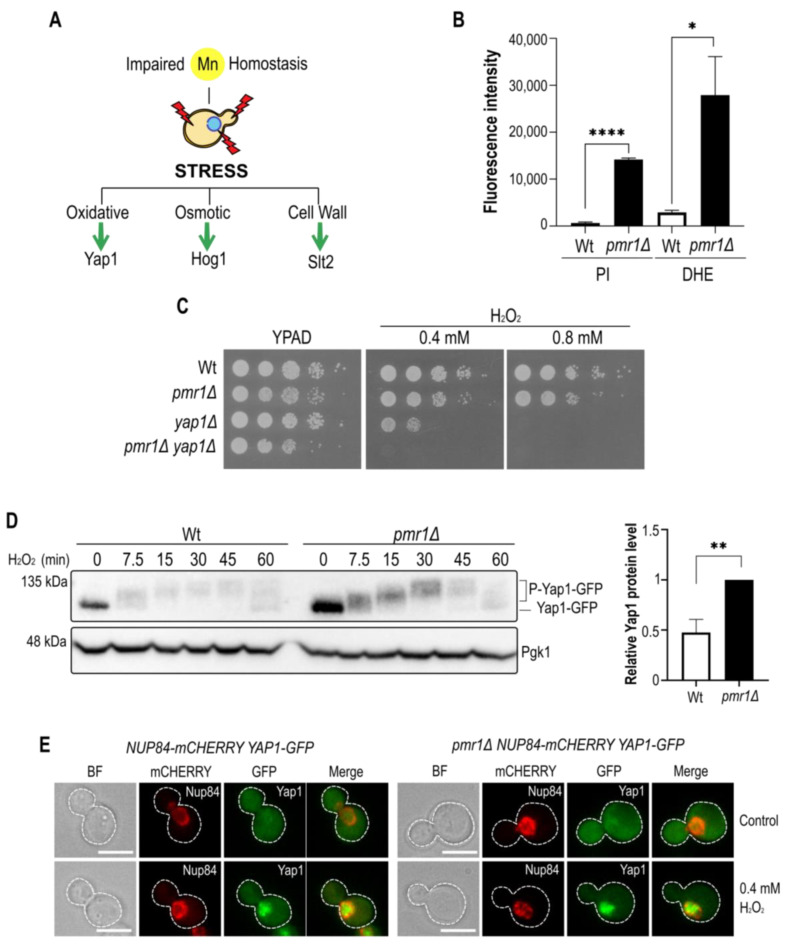
The manganese tolerance and oxidative stress resistance of Pmr1 depleted cells depends on Yap1. (**A**) The schematic outline of stress signaling pathways linked to oxidative stress. The effector transcription factor (Yap1) or MAPKs (Hog1, Slt2) are indicated. (**B**) Flow cytometric analysis of DHE and PI positive cells. The fluorescence intensity of positive Wt and *pmr1Δ* mutant cells is plotted. Standard error of the mean (SEM, with bars) and p-values are indicated (*p*-value 0.05 *; 0.01 **; 0.001 ****). (**C**) Growth on peroxide-containing (H_2_O_2_) YPAD (Yeast Extract-Peptone-Adenine-Dextrose) medium. 10-fold dilutions of exponentially growing cells are shown. Strains and H_2_O_2_ concentrations are indicated. (**D**) Immunoblot of Yap1-GFP modifications after H_2_O_2_ addition in Wt and *pmr1Δ* mutants using an anti-GFP antibody. H_2_O_2_ concentration and incubation times (min) prior to protein isolation are indicated. Pgk1 (3-phosphoglycerate kinase 1) protein levels, probed with anti-Pgk1 antibodies, served as a loading control. Quantification of relative Yap1-GFP levels in untreated Wt and *pmr1Δ* cells (right panel). Standard deviation (SD, with bars) and p-value are indicated. (**E**) Analysis of Yap1-GFP localization by fluorescence microscopy. Cells were analyzed before (control) and after 7.5 min treatment with 0.4 mM H_2_O_2_. Nup84-mCherry was used as a nuclear marker. Scale bar represents 5 µm.

**Figure 2 ijms-23-15706-f002:**
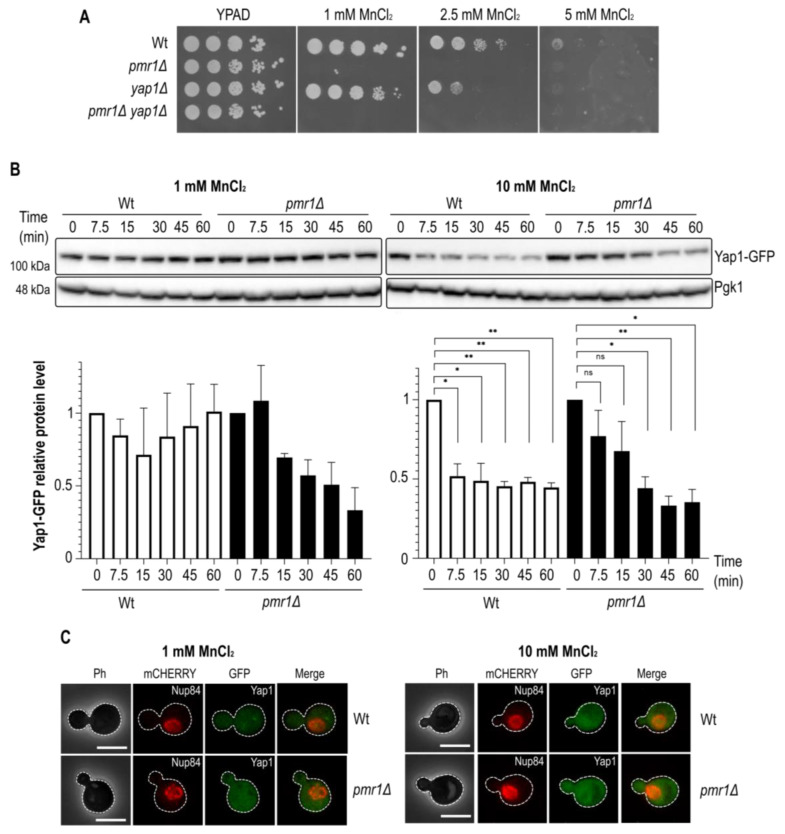
Yap1 is a target for manganese stress response. (**A**) Growth on MnCl_2_-containing medium. 10-fold dilutions of exponentially growing cells are shown. The strains and MnCl_2_ concentrations are indicated. The pictures were taken after 3 days of growth. (**B**) Immunoblot of Yap1-GFP protein levels after MnCl_2_ addition in Wt and *pmr1Δ* mutants. MnCl_2_ concentration and incubation times (min) prior to protein isolation are indicated. Pgk1 (3-phosphoglycerate kinase 1) protein levels, probed with anti-Pgk1 antibodies, served as a loading control. Quantification of Yap1 protein levels is shown below. SEM with bars and *p*-value are indicated (*p*-value 0.05 *; 0.01 **). Time point 0 was set to 1. (**C**) Fluorescence microscopy analysis of Yap1-GFP localization. The cells were analyzed before and after 15 min treatment with 1 or 10 mM MnCl_2_. Nup84-mCherry was used as a nuclear marker. Scale bar represents 5 µm.

**Figure 3 ijms-23-15706-f003:**
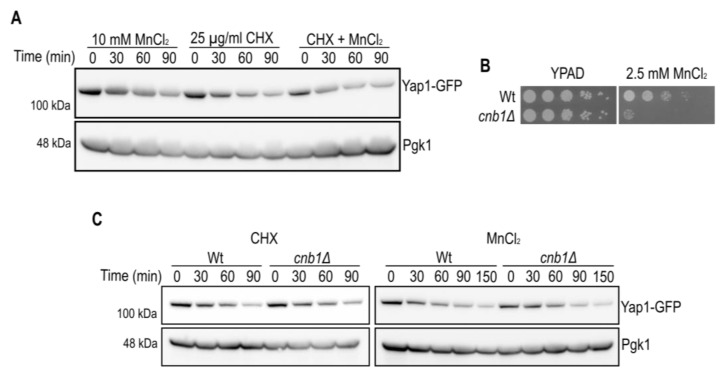
Calcineurin B is not involved in MnCl_2_-driven Yap1 decay. (**A**) Immunoblot of Yap1-GFP protein levels after MnCl_2_ and/or CHX addition. The incubation times (min) prior to cell collection are indicated. Pgk1 (3-phosphoglycerate kinase 1) protein levels served as a loading control. (**B**) Growth on MnCl_2_-containing medium. 10-fold dilutions of exponentially growing cells are shown. Strains and MnCl_2_ concentrations are indicated. The pictures were taken after 2 days of growth. (**C**) Immunoblot of Yap1-GFP protein levels in Wt and *cnb1Δ* mutant cells after CHX (**left** panel) or MnCl_2_ (**right** panel) addition. See (**A**) for further description.

**Figure 4 ijms-23-15706-f004:**
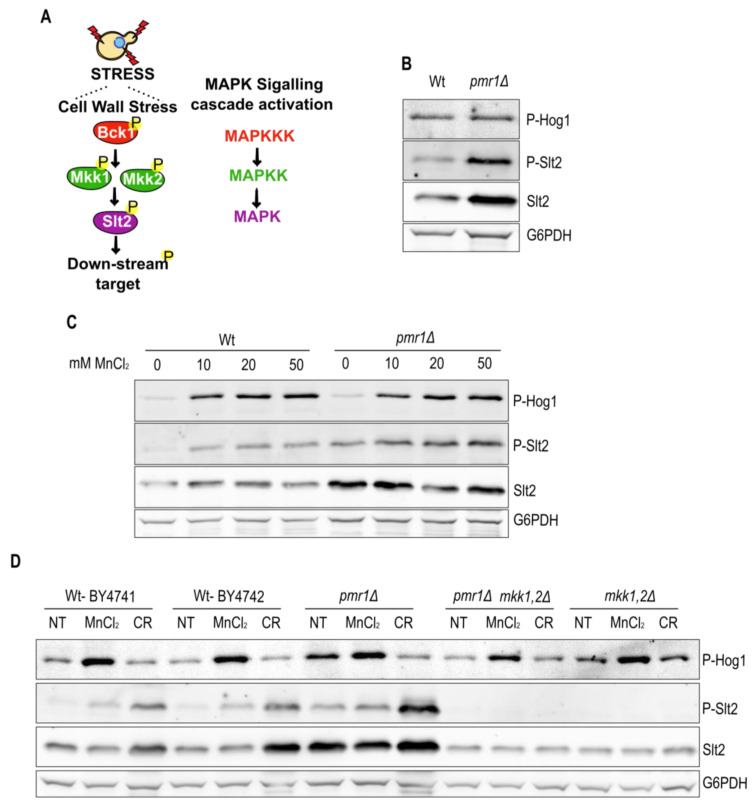
The constitutive Slt2 phosphorylation is a consequence of Pmr1 depletion. (**A**) The schematic outline of the MAPK cell wall integrity signaling cascade in yeast. The relevant kinases are shown. (**B**) Hog1 and Slt2 phosphorylation of exponentially growing Wt and *pmr1Δ* mutant were analyzed by immunoblotting using phospho-specific antibodies (anti-P-p38 MAPK Thr180/182 and anti-p42/44 MAPK Erk1/2, respectively). The total Slt2 levels were detected with anti-Slt2 antibodies. Glucose-6-phosphate dehydrogenase (G6PDH) levels were used as a loading control. (**C**) Analysis of Hog1 and Slt2 phosphorylation levels in cells treated for 1 h with different MnCl_2_ concentration. Description as in B. (**D**) Comparison of MnCl_2_-induced Hog1 and Slt2 phosphorylation in cells lacking the Slt2 upstream kinases Mkk1/2. The non-treated cells (NT) were used as negative control. Congo red (CR) treatment was used as a control to selectively induce Slt2 phosphorylation. Description as in (**B**).

**Figure 5 ijms-23-15706-f005:**
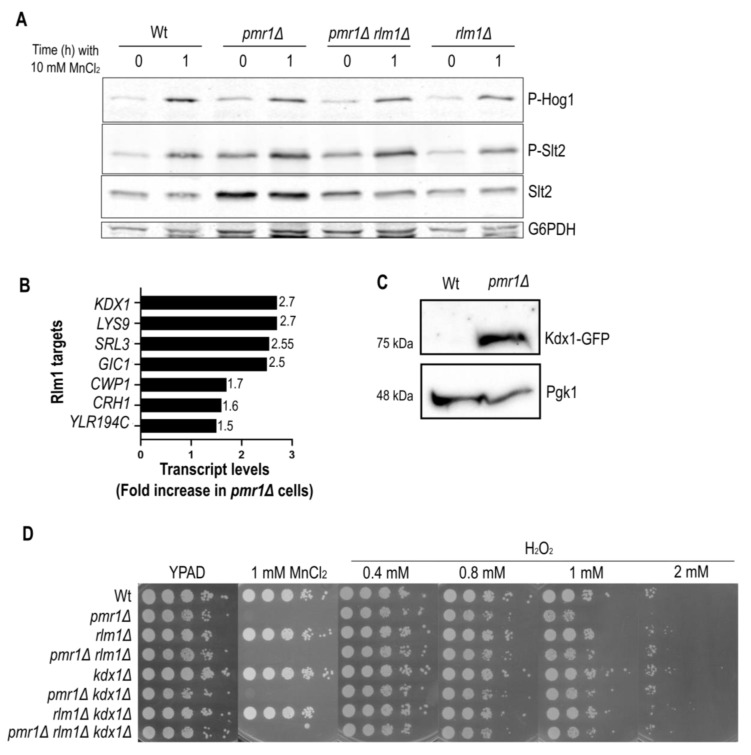
Slt2-downstream targets are up-regulated in cells lacking Pmr1. (**A**) Exponentially growing Wt and indicated mutant strains were treated for up to 1 h with 10 mM MnCl_2_. The levels of Slt2, phosphorylated Slt2 (P-Slt2) and Hog1 (P-Hog1) were analyzed by immunoblotting as in Figure 4. Glucose-6-phosphate dehydrogenase (G6PDH) was used as a loading control. (**B**) Up-regulation of Rlm1 target gene transcripts in *pmr1Δ* mutants and (**C**) Kdx1-GFP protein level, analyzed by immunoblotting with anti-GFP antibodies, in Wt and *pmr1Δ* mutants. Pgk1 was used as loading control. (**D**) The growth of 10-fold dilutions of exponentially growing cells is shown. Strains, MnCl_2_ and H_2_O_2_ concentrations are indicated. The pictures were taken after 2 days of growth.

**Figure 6 ijms-23-15706-f006:**
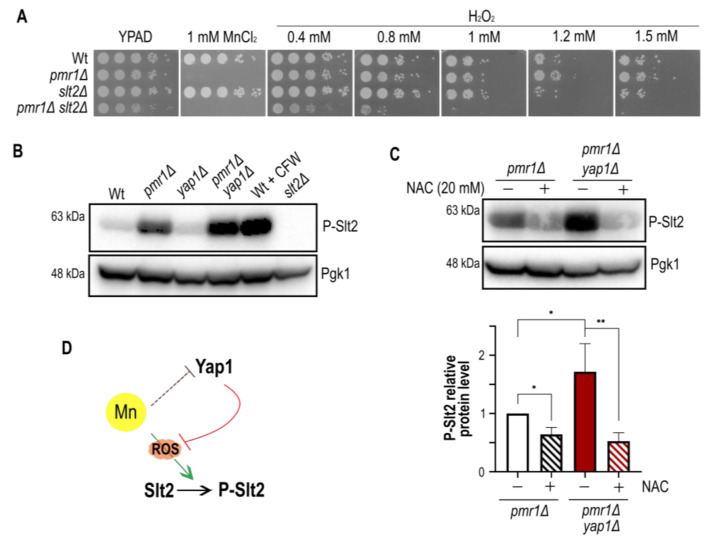
Slt2 has a dual role in ROS and manganese stress defense. (**A**) The growth of 10-fold dilutions of exponentially growing cells is shown. Strains, MnCl_2_ and H_2_O_2_ concentrations are indicated. Pictures were taken after 2 days of growth. (**B**) Comparison of Slt2 phosphorylation levels in cells lacking Pmr1 and/or Yap1. Wt cells treated with calcofluor white (CFW) and *slt2Δ* mutants were used as positive and negative controls, respectively. (**C**) Comparison of Slt2 phosphorylation levels in N-acetyl cystein (NAC) treated cells. The quantification of Slt2 phosphorylation is shown (bottom). SD (with bars) and p-value are indicated (*p*-value 0.05 *; 0.01 **). The value obtained for untreated *pmr1Δ* mutants was set as 1. (**D**) Model for manganese stress-induced Slt2 activation and Yap1 protein decay, respectively. See text for details.

**Table 1 ijms-23-15706-t001:** Yeast strains used in this work.

Strain	Genotype	Source
BY4741	MATa *ura3Δ0 leu2Δ0 his3Δ0 met15Δ0*	Euroscarf
BY4742	MATα *ura3Δ0 leu2Δ0 his3Δ0 lys2Δ0*	Euroscarf
YGL167C	MATa *ura3Δ0 leu2Δ0 his3Δ0 met15Δ0 pmr1Δ::kan*	Euroscarf
YHR030C	BY4741 *slt2Δ::kan*	Euroscarf
YKL161C	BY4741 *kdx1Δ::kan*	Euroscarf
YML007W	BY4741 *yap1Δ::kan*	Euroscarf
YPL089C	BY4741 *rlm1Δ::kan*	Euroscarf
YKL190W	BY4741 *cnb1Δ::kan*	Euroscarf
RWY160	MATa *ura3Δ0 leu2Δ0 his3Δ0 met15Δ0 NUP84mCHERRY::hph YAP1GFP::nat*	This study
RWY213	MATa *ura3Δ0 leu2Δ0 his3Δ0 met15Δ0 NUP84mCHERRY::hph YAP1GFP::nat pmr1Δ::kan*	This study
yWH1464	MATa *ura3Δ0 leu2Δ0 his3Δ0 met15Δ0 YAP1GFP::HIS3*	Woo-Hyun Chung
RWY258	MATa *ura3Δ0 leu2Δ0 his3Δ0 met15Δ0 YAP1GFP::HIS3 pmr1Δ::kan*	This study
RWY230	MATa *ura3Δ0 leu2Δ0 his3Δ0 met15Δ0 pmr1Δ::nat slt2Δ::kan*	This study
RWY223	MATa *ura3Δ0 leu2Δ0 his3Δ0 met15Δ0 pmr1Δ::kan kdx1Δ::hph*	This study
ASY003	MATa *ura3Δ0 leu2Δ0 his3Δ0 met15Δ0 pmr1Δ::nat rlm1Δ::kan*	This study
YMJ29	MATα *ura3Δ0 his3Δ0 lys2Δ0 mkk1Δ::S.p.HIS5 mkk2Δ::kan*	M. Molina
RWY219	MATα *ura3Δ0 his3Δ0 lys2Δ0 pmr1Δ::nat mkk1Δ::S.p.HIS5 mkk2Δ::kan*	This study
RWY225	MATa *ura3Δ0 leu2Δ0 his3Δ0 met15Δ0 pmr1Δ::nat yap1Δ::kan*	This study
RWY204	MATα *ura3Δ0 leu2Δ0 his3Δ0 lys2Δ0 KDX1GFP::nat*	This study
RWY208	MATα *ura3Δ0 leu2Δ0 his3Δ0 lys2Δ0 KDX1GFP::nat pmr1Δ::kan*	This study
RWY244	MATa *ura3Δ0 leu2Δ0 his3Δ0 met15Δ0 YAP1GFP::HIS3 rlm1Δ::kan*	This study
RWY246	MATa *ura3Δ0 leu2Δ0 his3Δ0 met15Δ0 YAP1GFP::HIS3 rlm1Δ::kan pmr1Δ::nat*	This study
RWY248	MATa *ura3Δ0 leu2Δ0 his3Δ0 met15Δ0 YAP1GFP::HIS3 slt2Δ::kan*	This study
RWY249	MATa *ura3Δ0 leu2Δ0 his3Δ0 met15Δ0 YAP1GFP::HIS3 slt2Δ::kan pmr1Δ::kan*	This study
RWY252	MATa *ura3Δ0 leu2Δ0 his3Δ0 met15Δ0 YAP1GFP::HIS3 kdx1Δ::kan*	This study
RWY253	MATa *ura3Δ0 leu2Δ0 his3Δ0 met15Δ0 YAP1GFP::HIS3 kdx1Δ::kan pmr1Δ::nat*	This study
RWY254	MATa *ura3Δ0 leu2Δ0 his3Δ0 met15Δ0 YAP1GFP::HIS3 rlm1Δ::kan kdx1Δ::hph*	This study
RWY256	MATa *ura3Δ0 leu2Δ0 his3Δ0 met15Δ0 lys2Δ0 YAP1GFP::HIS3 pmr1Δ::nat rlm1Δ::kan kdx1Δ::hph*	This study
RWY260	MATa *ura3Δ0 leu2Δ0 his3Δ0 met15Δ0 lys2Δ0 YAP1GFP::HIS3 rlm1Δ::kan slt2Δ::hph pmr1Δ::nat*	This study
RWY278	MATa *ura3Δ0 leu2Δ0 his3Δ0 met15Δ0 lys2Δ0 YAP1GFP::HIS3 cnb1Δ::kan*	This study

**Table 2 ijms-23-15706-t002:** Plasmids used in this work.

Plasmid	Description	Reference
pAG25-natMX4	DEL-MARKER-SET nourseothricin (nat)	Euroscarf
pFA6a-kanMX4	DEL-MARKER-SET geneticin G418 (kan)	Euroscarf
pAG32-hygMX4	DEL-MARKER-SET hygromycin (hph)	Euroscarf
GFP tagging	pYM-GFP-natNT2	Helle Ulrich

## Data Availability

The row data of the experiments shown in this study are available from the corresponding author upon request. The microarray data is openly available in the GEO database (www.ncbi.nlm.nih.gov/geo) under accession number GSE29420.

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
