# Peer review of "Manganese Stress Tolerance Depends on Yap1 and Stress-Activated MAP Kinases"

_ijms, 2022, doi:10.3390/ijms232415706_

Round 1

Reviewer 1 Report

The manuscript „Manganese Stress Tolerance depends on Yap1 and stress-activated MAP Kinases“ is an original research investigating molecular mechanisms of magnanese stress response in budding yeast. The manuscript is well written and the presentation of data on figures is clear. All figures are supported by original images of blots. The subject of the resarch is relevant. I can recommend the manuscript for publication in IJMS.

Author Response

We would like to acknowledge the reviewer's positive comments. 

Reviewer 2 Report

In this manuscript, the authors describe about the crosstalk of stress signaling pathways in yeast and provide evidence that impaired manganese homeostasis leads to oxidative stress and that cellular tolerance to MnCl2 requires the Yap1 transcription factor. The results are mostly solid and well presented. Also, the manuscript is well written and will be interesting not only to yeast researchers but also general readers. Therefore, I recommend this manuscript for publication in International Journal of Molecular Sciences. 

I have several minor comments as listed below.  

The authors should define "YPAD" somewhere in the text.

Line 150: "Nup84mCherry" should be " Nup84-mCherry ".

Lines 294 and 295: "MnCl2" should be"MnCl2".

Figure 1B: the condition should be described. Cells were treated with H2O2?

Figure 1D graph: the authors should describe how many minutes of processing time the graph is.

Figure 1E: hard to see the scale bars.

Table 1: the notation of null mutations should be unified. For example, "pmr1:kan" or "pmr1∆:kan". 

Table 1 YMJ29 and RWI219: "mkk" is "mkk1"?

Author Response

We would like to acknowledge the positive comments of the reviewer and his/her careful revision of this manuscript.
